# Markets as drivers of selection for highly virulent poultry pathogens

Justin K. Sheen [1] ✉, Fidisoa Rasambainarivo[1,2], Chadi M. Saad-Roy [3,4], Bryan T. Grenfell [1,5] & C. Jessica E. Metcalf[1,5]

Theoretical models have successfully predicted the evolution of poultry pathogen virulence in industrialized farm contexts of broiler chicken populations. Whether there are ecological factors specific to more traditional rural farming that affect virulence is an open question. Within non-industrialized farming networks, live bird markets are known to be hotspots of transmission, but whether they could shift selection pressures on the evolution of poultry pathogen virulence has not been addressed. Here, we revisit predictions for the evolution of virulence for viral poultry pathogens, such as Newcastle's disease virus, Marek's disease virus, and influenza virus, H5N1, using a compartmental model that represents transmission in rural markets. We show that both the higher turnover rate and higher environmental persistence in markets relative to farms could select for higher optimal virulence strategies. In contrast to theoretical results modeling industrialized poultry farms, we find that cleaning could also select for decreased virulence in the live poultry market setting. Additionally, we predict that more virulent strategies selected in markets could circulate solely within poultry located in markets. Thus, we recommend the close monitoring of markets not only as hotspots of transmission, but as potential sources of more virulent strains of poultry pathogens.

Marek's disease virus (MDV), Newcastle disease virus (NDV), and influenza virus, H5N1 are three of the deadliest pathogens that circulate in poultry populations. Since their discovery, all three pathogens have been shown to circulate worldwide, and the most virulent strains of each have been reported to kill up to 100% of poultry flocks[1-4]. These high rates of mortality, alongside the high frequency of disease outbreaks, make all three viruses major barriers to the sustainability of poultry as both food and economic resources in both industrialized and less-industrialized communities[5-8]. In rural settings, where households may be heavily reliant on their own backyard poultry, the nutritional and economic costs for any one household can be especially severe[6,9,10]. Understanding what processes might shape disease severity in such contexts thus has considerable applied importance.

Although rural settings remain under-studied, over the past two decades, an array of research has evaluated how industrialized farming practices might shape the evolution of 'virulence,' here defined as the rate of disease-associated host death. When higher virulence is positively associated with increasing rates of transmission, a tradeoff between transmission and mortality emerges. Although there is a benefit to increasing virulence due to increasing transmission, there is also a cost of shortening the lifespan of the host, and consequently the infectious period of the pathogen[11]. In the past, researchers have identified several husbandry practices that either increase the transmission benefits of virulence, or reduce the mortality costs of virulence ultimately selecting for increased virulence. These practices include vaccination with 'imperfect' vaccines, dramatic decreases in host lifespan in modern broiler poultry populations to fifty days or less, and cleaning between cohorts of poultry that does not completely eliminate the pathogen[1,11-15].

A natural question that arises is what ecological factors of rural settings may shape the evolution of virulence. The majority of the

[1]Department of Ecology and Evolutionary Biology, Princeton University, Princeton, NJ, USA. [2]Mahaliana Labs SARL, Antananarivo, Madagascar. [3]Miller Institute for Basic Research in Science, University of California, Berkeley, CA, USA. [4]Department of Integrative Biology, University of California, Berkeley, CA, USA. [5]School of Public and International Affairs, Princeton University, Princeton, NJ, USA. ✉e-mail: jsheen@princeton.edu

poultry population in rural localities is located in smallholder farms of primarily indigenous poultry[16]. Within this agricultural setting, live poultry may migrate from farms to regional live-bird markets through trade, and vice-versa, creating a spatially variable environment for poultry pathogens. Regional live-bird markets are hotspots for poultry pathogen epidemics[8,17,18]. They have been positively associated with the frequency of disease outbreaks, and consistently have higher prevalences than farm populations[9,19–24]. Because there appears to be uninterrupted pathogen transmission in markets (rather than extinction/reintroduction) markets can introduce selection pressures on the pathogens[9,25].

There are two reasons to expect that the selection pressures introduced by live bird markets will favor higher virulence. First, high rates of turnover and slaughter in markets may lower the average lifespan of poultry, which could select for increases in virulence, as indicated in broiler populations[13]. Second, high persistence of viral particles (environmental persistence) in markets could both increase viral transmission and create a reservoir of the virus, potentially increasing pathogen persistence in markets relative to farms[9,26,27]. Such increased environmental persistence also favors more virulent strategies, as reduced transmission time is offset by time in the environmental reservoir[26].

However, while many lines of evidence suggest that markets might select for increases in virulence, these hypotheses have yet to be rigorously examined. The magnitude of possible increases also remains unclear. Building on a rich literature on virulence evolution, we formalize expectations for selection pressures in markets (Fig. 1), explicitly grounding the theory in real-world parameters. We develop an epidemiological compartmental model to reflect pathogen transmission within markets (Fig. 2), and derive the $R_0$ (or number of secondary infections from an initial infection in a completely susceptible population) and calculate it for each virulence strategy. For our model, the global evolutionarily stable strategy (ESS) of virulence, or strategy that cannot be invaded by any other strategy, maps to the optimal

virulence strategy; this corresponds to maximizing the $R_0$ of the system relative to all other strategies (Appendix 2)[28–30]. Thus, we use the $R_0$ expression to titrate the impact of turnover rate and environmental persistence on the global ESS.

Our results indicate that across a broad range of parameter space, the higher turnover rate and higher persistence of viral particles in markets will select for higher virulence strategies. If appropriate mutations arise, markets can act as a source population for higher virulence strategies across the larger population. Increased cleaning to prevent environmental persistence of viral particles could play an important role in mitigating this outcome, although fast turnover rate may still dominate the selection pressures found in markets. Finally, we describe several possible extensions of our model, as well as which sources of empirical work could shed light on selection pressures on virulence in markets.

## Results

As expected, increases in both the turnover rate in markets, *m*, or cleaning in markets, *κ*, decrease the $R_0$ of all virulence strategies; the former because both the infected and susceptible poultry are turned over from markets at a faster rate, and the latter because of increased clearing of the environmental reservoir.

When both *m* and *κ* are allowed to decrease and increase, respectively, effectively changing market conditions to farm conditions, this decreases the global ESS for many, if not the majority of tradeoff curves for all values of $\Phi = 0.1, 1, 10,$ or 100 (Table S2, Appendix 3). When $\Phi = 1$, we find that global ESS decreases when we both decrease *m* and increase *κ* (78% have decreases in the global ESS, 22% have 0 change, maximum decrease is 499), as well as when *m* solely decreases (72% have decreases in the global ESS, 28% have 0 change, maximum decrease is 474), or solely *κ* increases (72% have decreases in the global ESS, 28% have 0 change, maximum decrease is 81). There were no tradeoff curves where global ESS increased as a result of decreasing *m* or increasing *κ*. Increased cleaning selects for

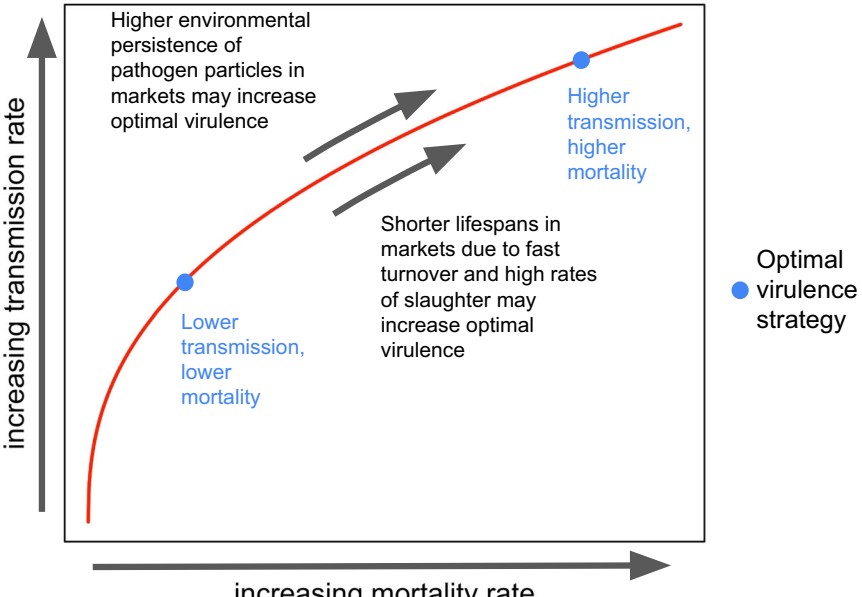

**Fig. 1 | Conceptual diagram of selection pressures on the evolution of the optimal virulence strategy of a poultry pathogen within markets.** Assuming a virulence transmission trade-off, where any increases in the mortality rate (virulence) of a poultry pathogen (x axis) is associated with an increase in the transmission rate of the poultry pathogen (y axis), following a concave trade-off curve (red line), various selection pressures cause different optimal virulence strategies to evolve. Lower and higher virulence strategies, corresponding to lower and higher mortality rates (x axis) and transmission rates (y axis) are shown in blue. Transmission in markets may increase the optimal virulence strategy due to higher rates of slaughter, and synonymously shorter lifespans, in markets. The higher environmental persistence of pathogens in markets may also increase the optimal virulence strategy. The optimal virulence strategy in our study is synonymous with the global evolutionarily stable strategy (ESS) of virulence.

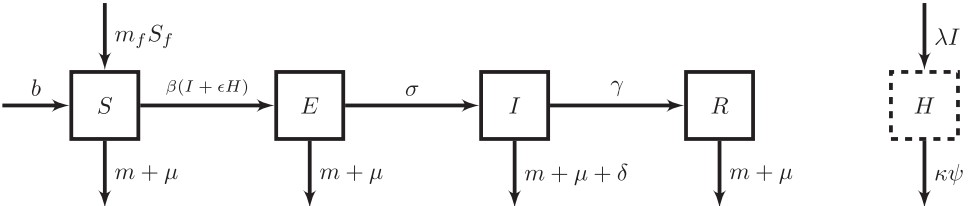

**Fig. 2 | Model for poultry pathogen transmission in markets.** indicating parameters that govern the inflow and outflow from each compartment. $N$ is the number of poultry in markets, $S$ is the number of susceptible poultry in markets, $E$ is the number of exposed but not infectious poultry in markets, $I$ is the number of infectious poultry in markets, $R$ is the number of recovered poultry in markets, $H$ is the number of units of reservoir in markets, $S_f$ is the number of susceptible poultry in farms, $\sigma = 1$/average incubation period of the poultry pathogen, $\gamma = 1$/average infectious period of the pathogen, $\delta$ = per capita per day disease-related death rate of poultry, $\beta$ = per day transmission rate of pathogen in markets, $m_f$ = per capita per day rate of migration of poultry from farms into markets, $m$ = per capita per day rate of migration of poultry out of markets, $b$ = background rate of birth, $\mu$ = background rate of natural death, $\lambda$ = per capita shedding rate of pathogen particles to the environmental reservoir, $\Psi$ = baseline per capita per day rate of clearance of environmental reservoir through natural pathogen decay, $\kappa$ = multiplicative factor of baseline rate of clearance of environmental reservoir due to cleaning, and $\varepsilon$ = relative efficacy of environmental transmission vs. contact transmission. The solid-line compartments denote compartments where the unit is of poultry, while the dashed-line compartment denotes the compartment where the unit is environmental pathogen particles.

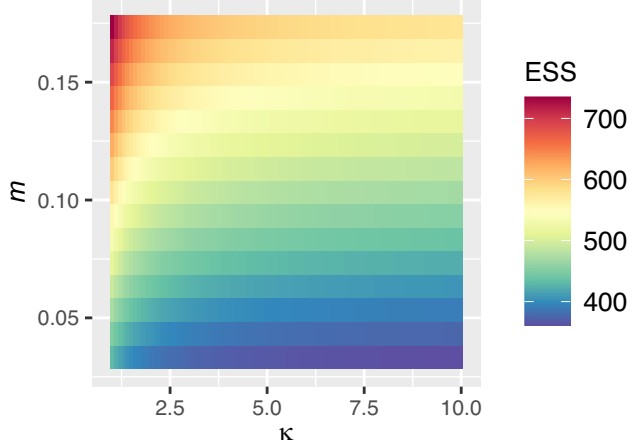

**Fig. 3 | Faster turnover rate ($m$) and higher environmental persistence (proportional to $1/\kappa$) in markets relative to farms can select for a higher optimal virulence strategy (global ESS).** The precise rate of environmental persistence is not known in any setting, but market conditions are likely to correspond to higher persistence and low $\kappa$ (left of plot) relative to farm conditions (right of plot); and likewise markets are likely to have higher turnover $m$ (top of plot) than farms (see "Table S1: Relative comparisons of key characteristics in farms vs. markets that may shape selection pressures for the evolution of virulence"). Tradeoff curve parameters used are $c_1 = 1/2300$, $c_2 = 0.6$ and $\Phi = 1$.

a lower global ESS since environmental transmission benefits will decrease to zero with more cleaning. Thus, the higher virulence strategies that stand to gain the most from environmental transmission when there is no cleaning (since $\lambda$ is an increasing function of virulence) will suffer the greatest reduction in transmission benefits when there is cleaning. Without this added environmental transmission benefit, higher virulence strategies are less valuable from the pathogen's perspective, and lower virulence strategies are selected.

When there was differential turnover of infectious poultry compared to all other poultry, i.e., infectious poultry emigrate at rate $m_I = 50\%$ slower than $m$, while the global ESS still decreases as a result of decreasing $m$, they decrease at smaller amounts for all values of $\Phi = 0.1, 1, 10,$ or $100$ (Table S3, Appendix 3). Thus, when infectious poultry emigrate at slower rates, this minimizes the magnitude of the decrease in the global ESS. Preliminary results from our sensitivity analysis assuming distinct cohorts periods, with cleaning solely between cohorts, suggests that in these scenarios, cleaning will instead increase virulence (Appendix 6).

Differences in the optimal virulence strategy in farms and markets will depend solely on differences in the parameters $m$ and $\kappa$ (see "Methods: Characterizing ecological factors of markets"). Thus, because farms have relatively lower m and relatively higher $\kappa$, markets will have higher optimal virulence strategies than farms (Fig. 3).

## Discussion

Pathogen evolution can have devastating impacts on livelihoods and sustainability[13]. Live bird markets are known to shape prevalence and persistence of poultry pathogens indicating that they may also affect poultry pathogen evolution. Our analysis shows that markets could select for and sustain higher virulence strategies relative to farms; this is due to both the fast turnover rate of poultry and higher environmental persistence of pathogens on markets relative to farms. The former echoes a classic result in the literature of virulence evolution where higher host mortality will select for increased virulence, but in the novel context of live poultry markets[11,13,22]. The exact increase in virulence will depend on the specific transmission-mortality tradeoff curve, but in some cases, the optimal virulence strategy selected increases to more than twice the baseline virulence, infecting over 50% as many susceptible poultry per infected poultry per day. While we use NDV-specific parameters of transmission, many poultry viruses such as MDV and H5N1 are found in live bird markets and may consequently undergo similar selection pressures for higher virulence[31,32]. Clearance of the pathogen from the environment may not only decrease the prevalence of these poultry pathogens, but lessen the selection pressure for higher virulence.

Our results suggest considerable returns on investment into surveillance of pathogens in markets, both in characterizing the impact of mitigation strategies, such as cleaning, on pathogen prevalence, as well as tracking trajectories of pathogen virulence. Viral genetic surveillance may be especially valuable to identify whether high virulent strains circulate in markets, given the known virulence factors for pathogens such as NDV and H5N1, or identify new ones, especially if these may be complex and multi-site specific, such as in the case for MDV[15,33,34]. Future transmission models that take into account spatial complexities such as farm-to-market or market-to-market connections can help inform where to sample viruses for optimal surveillance.

Our result that cleaning decreases the global ESS contrasts with a previous result, finding that cleaning could increase virulence for industrialized poultry populations[1]. Crucially, the previous result assumes that all poultry of one cohort are removed from the system before the next cohort arrives i.e., all-in-all-out dynamics. Our sensitivity analysis suggests that if we assume this dynamic, with cleaning solely during the intercohort period, cleaning will increase the global ESS (Appendix 6). However, these all-in-all-out dynamics may be less

common in the rural live-poultry markets we model. Sellers often receive a new cohort of poultry to sell before all of the poultry of the older cohort are sold. In addition, even if some sellers may practice all-in-all-out cohort practices, the number of infectious poultry of each new cohort of one seller may depend not only on how well the seller cleans the particles from their own stall, but also on the prevalence in markets. Thus, the continuous dynamics we model may be more appropriate for modeling rural market systems.

Extensions to our model incorporating other differences between farms and markets may create even larger differences in virulence. Small farms, usually averaging fifteen poultry, are often also less connected than the contiguous live-bird trade. This will reduce the effective community size for pathogen circulation, and increase prospects for stochastic pathogen extinction, further decreasing the population-wide optimal strategy of virulence in farm communities in comparison to markets[35–37]. While we do allow for the probable higher contact rate of birds in markets than farms, this scales $R_0$, eliminating effects on the evolution of virulence[38]. In models with explicit spatial structure, the higher contact rate may also lead to higher virulence, as above, because the effective population size is larger. Free-roaming poultry on farms may have the opportunity of encountering poultry from neighboring farms, but this still might not exceed the range of contacts of poultry in markets, due to the crowded transportation of poultry from as many as 80 farms or 20 villages overnight[18]. Empirical measurements such as surveys to identify the local contact network of poultry and contact rates in market systems would shed light on the importance of these potential model extensions.

Theory can provide a basis for why we should expect live bird markets to select for higher virulence, but whether or not we observe differences in virulence across the farm-market divide will ultimately depend on real-world details specific to the live bird market sampled. More specifically, if transmission is contained within farms and markets, separate evolutionarily stable strategies will evolve, with farm communities favoring evolution of less virulent pathogens via the slower turnover rate and higher cleaning rate, with occasional spillover of highly virulent strains from markets. But if there is strong connectivity between farms and markets, additional selection pressures of farms would have to be considered in addition to those solely found in markets. Phylogenetic clustering of isolates in markets vs. farms supports the existence of separate 'farm' and 'market' niches[9]. Consistently higher prevalence in markets compared to farms also suggests a market-specific niche, characterized by transmission pathways unique to markets[9,19,23,24]. Since markets are normally a terminal endpoint in the lifespan of poultry, and consequently the pathogen itself, pathogen strains markets will rarely return to farms and will primarily experience selection pressures in markets for higher virulence[18]. The actual virulence trade-off curve of the pathogen will also play a role in how large the difference should be. Thus, detecting transmission overlap between farms and markets is an important direction for research.

The rapid turnover rate in live-poultry markets will select for higher virulence, but additional factors, such as multiple infections, also remain an important area of future research. When hosts have multiple infections, there is higher within-host competition, which selects for higher virulence[39,40]. When there is higher background host-mortality, within-host competition is potentially lessened since there is a lower force of infection, and lower virulence may be selected. Thus, when there are multiple infections, this may potentially change our result that rapid turnover, and higher background host mortality, will select for increased virulence. Because poultry viruses such as Newcastle disease virus are known to have many coinfections[41], this possibility should be considered in future research, specifically whether the higher background host mortality in live-poultry markets would reduce the force of infection of multiple pathogens enough to select for lower virulence.

Given these causes for caution around markets, not only as sources of viral persistence, but as sources for more virulent viruses,

there are also likely to be high returns on investment in cleaning. The exact benefit will depend on the efficacy of environmental transmission, the clearance time of the environmental reservoir, how many viral particles are excreted per individual, as well as the true virulence tradeoff curves[1,13,42]. While increases in virulence still occur when turnover rate is fast, cleaning induces an opposing selection pressure that prevents the selection of highly virulent strategies (Fig. 3). The implementation of rest days suggested previously may also not only slow transmission in markets, but also prevents higher virulence[22]. However, more recent work has shown how time-varying trends in the susceptible population, a possibility in these live-bird markets due to potential seasonality, may add further complexity in our predictions of observed virulence in the short and long-term[43]. The consideration of these evolutionary consequences becomes more urgent as the poultry of live-bird markets are increasingly used as a sustainable food and economic resource.

## Methods

Our aim is to characterize the evolution of the optimal level of virulence, or host mortality rate, in contexts where there is a relationship between virulence and transmission (Fig. 1), when transmission occurs in markets. To do this, we perform an adaptive dynamics analysis to find how the global evolutionarily stable strategy (ESS) changes under market conditions. We first derive and analyze the $R_0$ of a transmission model designed to capture the market conditions of faster turnover rates and higher environmental persistence of viral particles in markets ("Transmission assumptions in markets"). We then define our assumptions of the transmission-mortality tradeoff curve ("Virulence assumptions"). Finally, we decrease the turnover rate and increase the cleaning parameters in order to vary the parameters from market conditions to farm conditions and evaluate how the global ESS changes ("Characterizing ecological factors of markets"). The code used to generate these results, as well as the results themselves, are openly available at http://www.github.com/jsheen/marketVirEvol[44].

### Transmission assumptions in markets

Our model focuses on the infection dynamics within markets, making a simplifying assumption of a constant number of susceptible poultry that migrate from farms to markets. We use a modified version of the susceptible-exposed-infected-recovered (SEIR) model that additionally includes migration into and out of the system, as well as a compartment to model persistence of the environmental reservoir of the pathogen (Fig. 2). The SEI model framework has been selected as an appropriate model for poultry pathogens such as Newcastle disease virus and Marek's disease virus[45,46]. Although in the current model, recovered individuals do not affect infection dynamics, we extend the SEI model to include a recovered class to model the possibility of recovered individuals affecting infection dynamics in future extensions (e.g., when transmission is frequency-dependent). The equations of the model are:

$$\frac{dS}{dt} = b - \beta SI - \varepsilon\beta SH - \mu S + m_f S_f - mS \tag{1a}$$

$$\frac{dE}{dt} = \beta SI + \varepsilon\beta SH - \sigma E - \mu E - mE \tag{1b}$$

$$\frac{dI}{dt} = \sigma E - mI - \gamma I - \delta I - \mu I \tag{1c}$$

$$\frac{dR}{dt} = \gamma I - mR - \mu R \tag{1d}$$

$$\frac{dH}{dt} = \lambda I - \kappa\psi H \tag{1e}$$

**Table 1 | Parameters of the transmission model for poultry pathogen transmission in markets**

| Parameter | Description | Value | Reference |
|---|---|---|---|
| $\sigma$ | 1/average incubation period of the pathogen | 1/5 days | Chukwudi et al.[55] |
| $\gamma$ | 1/average infectious period of the pathogen | 1/5 days | Tatár-Kis[56]. |
| $m_f$ | Per capita per day rate of migration of poultry from farms into markets | 10%/4 months | Assumed |
| $m$ | Per capita per day rate of migration of poultry out of markets | 1/5.5 days | Data collected in Madagascar |
| $b$ | Birth rate of poultry to raise in markets | 0 | Assumed to be negligible |
| $\mu$ | Background rate of natural death | 1/365 days | Omiti et al.[49] |
| $\varepsilon$ | Relative efficacy of environmental transmission vs. transmission through contact | 1 | Assumed |
| $\Psi$ | Baseline per capita per day rate of elimination of environmental reservoir through natural pathogen decay | 1/5 days | Assumed to be the same as the average infectious period of an infectious poultry |
| $\kappa$ | Multiplicative factor of baseline rate of clearance of environmental reservoir due to cleaning | 1 | Assumed that cleaning normally does not occur |
| $\beta$ | Per day transmission rate of pathogen in markets | $c_1(\alpha)^{c_2}$ | Set according to virulence strategy |
| $\delta$ | Per capita per day disease-related death rate of poultry | $c_3(\alpha)$ | Set according to virulence strategy |
| $\lambda$ | Per capita per day shedding rate of viral particles into the environmental reservoir | $\Phi c_1(\alpha)^{c_2}$ | Set according to virulence strategy |

Defined using the epidemiological and biological parameters of Newcastle disease virus. For data collected on the rate of emigration of poultry from live-poultry markets, *m*, we calculate the mean number of days until poultry were sold across 130 market sellers (3.5 days) plus an additional two days to account for transit time by middlemen (see "Methods: Characterizing ecological factors of markets").

Where $N$ (which equals $S + E + I + R$) is the total number of poultry in markets, $S$ is the number of susceptible poultry in markets, $E$ is the number of exposed but not infectious poultry in markets, $I$ is the number of infectious poultry in markets, $R$ is the number of recovered poultry in markets, H is the units of pathogen particles in the environmental reservoir, $S_f$ is the number of susceptible poultry in farms which we assume is a constant, $\sigma = $ 1/average incubation period of the poultry pathogen, $\gamma = $ 1/average infectious period of the pathogen, $\delta = $ per capita per day disease-related death rate of poultry, $\beta = $ per day transmission rate of pathogen in markets, $m_f = $ per capita per day rate of migration of poultry from farms into markets, $m = $ per capita per day rate of migration of poultry out of markets, $b = $ background rate of birth, $\mu = $ background rate of natural death, $\lambda = $ per capita shedding rate of pathogen particles into the environmental reservoir per infectious poultry, $\Psi = $ baseline per capita per day rate of clearance of environmental reservoir through natural pathogen decay, $\kappa = $ multiplicative factor of baseline rate of clearance of environmental reservoir due to cleaning, and $\varepsilon = $ relative efficacy of environmental transmission vs. transmission through contact. The values for all parameters are described in Table 1. Conceptual differences in husbandry practices in markets compared to farms are described in Table S1. In this article, we model density-dependent transmission of both the infectious poultry and environmental reservoir: assuming the area of the markets modeled is fixed, a larger population size will increase the contact rate of poultry between poultry or with the environment; thus, the contact rate is density-dependent. In our model $\varepsilon$ is set to 1, but we keep $\varepsilon$ in the model equations for future extensions.

Although poultry are primarily located in live-bird markets, our model accounts for times when poultry are transported by middlemen to traders, as well as if they are housed anywhere on off-days of markets. As long as poultry are continuously mixing in these places, our model is appropriate for modeling pathogen transmission (e.g., settings like Antananarivo, Madagascar where markets are open year-round). Vaccination is rare, and even when used, irregular, in many of these rural localities in both farms and markets[9,47].

Although we expect negligible hatching in markets, meaning that the birth rate, b, should be approximately zero, we include a term for births in our model equations in order to show that even with births, this should not influence our evolutionary results (see "Justification of comparison of selection pressures in markets vs. farms" in Appendix 3). We assume that recovered poultry will never become

susceptible or infectious again, which is a reasonable assumption due to the high antibody response to infection and relatively short life-cycle of the host (~1 year)[48,49]. Infectious poultry outside of the markets, i.e., in farms, cannot transmit to poultry in the markets, and vice-versa. We assume dead poultry are removed and thus make negligible contributions to transmission.

The $R_0$ of our model of transmission in markets, derived in Appendix 1, is:

$$R_0 = \left(\frac{m_f S_f}{\mu + m}\right)\left(\frac{\sigma}{\sigma + \mu + m}\right)\left(\frac{1}{m + \gamma + \delta + \mu}\right)\beta\left[1 + \frac{\epsilon\lambda}{\kappa\psi}\right] \quad (2)$$

The $R_0$ factors and terms can be interpreted biologically. $\sigma/(\sigma + \mu + m)$ is the probability an exposed individual will survive the incubation period without migration and death. Multiplying $1/(m_m + \gamma + \delta + \mu)$ through the transmission terms in brackets, $\beta/(m + \gamma + \delta + \mu)$ is the average transmission produced by an infectious individual; while $\epsilon\beta\lambda/\kappa\psi(m + \gamma + \delta + \mu)$ is the average transmission produced by a unit of the environmental reservoir of pathogen particles. $m_f S_f/(\mu + m)$ is the disease-free equilibrium population.

Several characteristics of selection pressures on $R_0$ emerge from this expression. First, because $\beta$ can be factored out of the equation, a higher contact rate in markets relative to farms will simply raise the $R_0$ of all virulence strategies by a constant, and thus will not change the position of the global ESS. This recapitulates a result found in van Baalen et al. for a simpler transmission model[38]. Similarly, the shape of the $R_0$ fitness landscape will be scaled by the migration rate from farms to markets ($m_f$) and the number of susceptible poultry in farms ($S_f$), and thus these factors will not affect selection pressures (see "Justification of comparison of selection pressures in markets vs. farms" in Appendix 3). We focus on the consequences of the two remaining ecological factors of markets: the fast turnover rate, and the higher persistence of viral particles in markets.

We find the optimal virulence strategy by identifying the virulence strategy with the largest $R_0$ in the system, i.e., following an adaptive dynamics approach[28]. In Appendix 2, we prove that the virulence strategy with the largest $R_0$ relative to all other virulence strategies: will be globally evolutionarily stable, will be able to invade all other virulence strategies, and will be convergently stable. Our model is a special case of a general transmission model developed for cholera which has a unique, stable, endemic equilibrium when $R_0 > 1$[50,51]. Thus, we have

shown that to perform an adaptive dynamics analysis we solely need to find the virulence strategy that maximizes $R_0$ relative to all other virulence strategies, since this strategy is the global evolutionarily stable strategy, and all virulence strategies have a unique endemic equilibrium. From another perspective, the effect of the environment can be characterized by a single resource, namely the equilibrium of susceptible individuals; hence, finding the virulence strategy that maximizes $R_0$, resulting in the lowest equilibrium of susceptibles, is sufficient for predictions of virulence evolution[52].

While we model density-dependent transmission, our results may hold in the frequency-dependent case. In particular, in Appendix 5, we prove that when transmission is frequency dependent there is a unique endemic equilibrium when $R_0 > 1$. If this equilibrium is stable and $R_0$ maximization is a valid approach for an adaptive dynamics analysis when transmission is frequency-dependent, then our results directly translate to the frequency-dependent setting.

Only susceptible poultry from farms are allowed to migrate to markets. Conceptually, more symptomatic poultry are less likely to be sold to middlemen and traders due to their diseased appearance. In addition, our adaptive dynamics analysis cannot account for when infected poultry from farms are able to migrate to markets, since in this case, it has not yet been proven that there is a unique endemic equilibrium for each virulence strategy. Future studies may aim to account for cases where infectious poultry can migrate to markets.

### Virulence assumptions
The level of virulence, $\alpha$, of a pathogen will influence both the rate of its transmission in markets, $\beta$, the rate of disease-associated mortality, $\delta$, as well as the shedding rate of environmental pathogen particles into a reservoir, $\lambda$, according to the following equations for a given virulence strategy:

$$\beta = c_1 \alpha^{c_2} \tag{3a}$$

$$\delta = c_3 \alpha \tag{3b}$$

$$\lambda = \Phi c_1 \alpha^{c_2} \tag{3c}$$

The first two equations for $\beta$ and $\delta$ create a concave-down tradeoff relationship between transmission and disease-induced mortality (as depicted in Fig. 1). Since the true trade-off curves for NDV and MDV are unknown, we explore a wide range of parameter space by calculating the transmission and disease-induced mortality rate for all virulence strategies, $\alpha$, between 0 and 1000 taking values of $c_1$ and $c_2$ that balance the flatness of the tradeoff as well as magnitude of transmission ("Appendix 3: Justification for range of $c_1$ and $c_2$"). $c_3$ is set at 1/1000 such that at the maximum virulence, the per capita of infectious poultry per day death rate is 1. A direct relationship between virulence and pathogen shedding has been shown to be a reasonable assumption, with past empirical and theoretical work indicating that higher virulence increases the within-host pathogen number, supporting our modeling assumption of $\lambda$[26]. We vary $\Phi$ from 0.1, to 1, to 10, to 100. To bound our results, we exclude all parameter combinations where the maximum transmission rate achieved of a given tradeoff curve is either less than one bird per day in a completely susceptible population, or greater than one hundred poultry per day in a completely susceptible population.

### Characterizing ecological factors of markets
To explore the impact of market conditions on selection on virulence, we decrease turnover rate ($m$) and decrease environmental persistence (by increasing clearing of viral particles parameter, $\kappa$) from the baseline of market conditions ($m = 1/5.5$ days; $\kappa = 1$) and check that (1) this reduces the $R_0$ of all strategies and (2) decreases the optimal virulence strategy, for all values of $\Phi = 0.1, 1, 10,$ or 100. Our analysis

compares the global ESS in farms vs. markets, since $m$ and $\kappa$ are lower and higher in farms compared to markets, respectively. We report the percentage of tradeoff curves where the global ESS is lowered, as well as the maximum amount the global ESS is lowered across all tradeoff curves, when we vary solely $m$ from 1/5.5 days to 1/365 days, when we vary solely $\kappa$ from 1 to 10, where larger $\kappa$ is equivalent to higher overall rate of clearance due to cleaning, and when both $m$ and $\kappa$ are decreased and increased, respectively. We exclude tradeoff curves that lead to an infectious poultry infecting more than 200 poultry per day in a completely susceptible population at maximum virulence, or less than 1 poultry per day in a completely susceptible population in the baseline market conditions. We also exclude those tradeoff curves where the global ESS under the baseline market conditions has an $R_0 > 100$ or $R_0 < 1$.

To inform the per capita per day rate of migration of poultry from farms into markets, $m_f$, we based ourselves on results from Kenya indicating that 337 to 1490 poultry are sold per week in a single town[18], that majority of which were from backyard farms. Since we are modeling a region made up of several towns (Table 1), we reasoned that our rate should be higher than this. If the population of poultry in farms is 1 million (Table S3), setting ~10% of poultry migrating from farms to markets in four months in a given region results in approximately 5833 poultry migrating to markets per week.

To inform the per capita per day rate of migration of poultry out of markets, $m$, we analyzed survey data collected from live-poultry market traders (47 from Toamasina, Madagascar and 83 from Antananarivo, Madagascar) of the average number of days poultry spend in markets before being sold. Market traders were asked to participate in the study, and those who opted-in were surveyed, making this a convenience sample. We found that the average number of days poultry are kept in markets before being sold among this sample was 3.5 days. We added 2 days to account for transit time with middlemen before migrating to markets, since most poultry are sold within 1 day by middlemen[53], though some middlemen keep their poultry longer—sometimes overnight[18], giving an average to upper-bound duration of 5.5 days that poultry stay with middlemen or traders (Table 1). The data on the average duration that poultry stay in live-poultry markets in this study have been deposited in the Zenodo database (see "Data availability"). Informed consent was obtained and approved by the Princeton University IRB office (IRB Record Number 12138).

Since it is possible that less-diseased poultry may be less likely to be sold and thus remain in markets for a longer period than other poultry, we perform a sensitivity test of our results when the emigration rate from markets is slower for infectious poultry (Appendix 4). We also perform a sensitivity analysis of our cleaning result when there are distinct cohorts, where all poultry of one cohort are replaced by a new cohort, with cleaning between cohorts as in industrialized poultry populations (Appendix 6).

### Reporting summary
Further information on research design is available in the Nature Portfolio Reporting Summary linked to this article.

## Data availability
The data on the average duration that poultry stay in live-poultry markets in this study have been deposited in the Zenodo database under accession code https://doi.org/10.5281/zenodo.10403195[54].

## Code availability
Simulated data, code, and results that support the findings of this study are openly available in GitHub at http://www.github.com/jsheen/marketVirEvol. These have also been deposited in the Zenodo database under accession code https://doi.org/10.5281/zenodo.10373146.44.

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

## Acknowledgements
C.J.E.M. and B.T.G. were supported by the Princeton Catalysis Initiative. C.M.S.R. acknowledges funding from the Miller Institute for Basic Research Science of UC Berkeley via a Miller Research Fellowship. The authors thank Christopher D. Golden for helpful comments concerning features of live-poultry markets.

## Author contributions
J.K.S., F.R., and C.J.E.M. designed the study. J.K.S. performed the simulations. F.R. collected data used in simulations. J.K.S. and C.M.S.R. performed the analyses. J.K.S., B.T.G., and C.J.E.M. wrote the initial draft of the article. All authors contributed equally in interpreting the results and editing the article.

## Competing interests
The authors declare no competing interests.
