## [Peer Review File · Nature Communications]

Markets as drivers of selection for highly virulent poultry pathogensREVIEWER COMMENTS

Reviewer #1 (Remarks to the Author):

In this manuscript, the authors construct a mathematical model with the aim of investigating virulence evolution in rural markets and farms. Overall, this is an interesting question – “what ecological factors in rural settings shape virulence evolution” - that has yet to be addressed with mathematical modelling.

I believe the authors have a bit more work to do. My major concern is with the evolutionary analysis. On line 203 the authors reference ‘The Hitchhiker’s Guide to Adaptive Dynamics’. In this reference the evolutionary stable strategy is found via optimizing invasion fitness. But what the authors have done in this manuscript is optimize R_0 . I suggest the authors calculate invasion fitness in this study, or explain why optimizing R_0 is the better approach.

Please see ‘Beyond R_0 Maximisation: On Pathogen Evolution and Environmental Dimensions’ by Lion and Metz. <https://pubmed.ncbi.nlm.nih.gov/29665966/> for clarification.

As the authors revisit the mathematical analysis of this model, I think it would be interesting to explore a “migration out of markets” that is dependent of disease status. I would think that diseased birds (and since they are not vaccinated, it is likely they would look “sick”) would “migrate” out of markets at a slower rate.. since they look less delicious. This would be interesting since this could be a negative selective pressure on virulence evolution. The lifespan of infected birds would be longer for “sick” birds, and this would select for less virulent varieties.

There is a typo in the model presented in the main text. There is no ‘p’ term to toggle between FD and DD transmission. Some motivation for FD vs DD transmission would be helpful. When comparing Newcastle, Marek’s and HPAI, are one of those viruses assumed to spread via FD transmission?

It would also be helpful if the authors could elaborate on what the reservoir, ‘ H_m ’, represents biologically. At first, I assumed this was a free living “viral” population. You mention on line 168 that “mucus and faeces are continually shed into the environmental reservoir for each infectious poultry in the system.” But, there is no contribution to this ‘ H_m ’ compartment from the infectious class (only the exposed class). So, it is unclear what this compartment represents.

I think if the authors make these changes, this can be a very interesting and worthwhile paper.

Reviewer #2 (Remarks to the Author):

This study explores the effect of rural markets on the evolution of virulence of poultry pathogens. The spread of the pathogen in a market is governed by the influx and the outflux of poultry and by the transmission between birds but also form an environmental compartment. The pathogen population is assumed to be unaffected by evolution going on in other farms or other markets. The analysis is based on the optimisation of R_0 : the ESS virulence is the one maximizing R_0 but this value may vary with the parameters that govern the life cycle in the market. The authors conclude that ESS virulence increases with fast turnover and less cleaning.

I have several major issues with this work:

(1) Some assumptions of the model are unclear. H_m is supposed to be “units of reservoir in markets” but this probably means the density of pathogen propagules in the environment. Why are these propagules produced by E_m at a rate σ (E_m are supposed to be not infectious)? Also, why do you introduce β_m as a function of H_m ? It would be simpler to model the density H_m to be a function of infectious hosts (i.e. the first term of the last equation could be $\beta_m I_m$, where the E would stand for the production of propagules in the environment. In addition, the parameter p is not well defined. I understand it affects R_0 but it should be

introduced in the EDO model too.

(2) The prediction that increased mortality selects for higher ESS virulence is a very classic prediction (a prediction that has been put to the test experimentally some time ago, Ebert and Mangin 1997 in *Evolution*). Increasing turnover is the same effect and it should be discussed.

(3) The prediction that less cleaning should select for more virulence contrast with previous analysis. In ref [1] the authors show that increased cleaning is expected to select for higher virulence. Why the difference? Probably in the underlying assumption of the demography. The present work assumes the population has reached a stable endemic equilibrium before the introduction of a new mutant. In ref [1] the demography fluctuates periodically. None of these assumptions are likely to be relevant in a market so it is difficult to make predictions about the effect of cleaning.

(4) More generally, I am not sure the looking for the ESS is most relevant approach to this question as we may want to determine the fate of a mutant during a transient epidemic. Different tools have been used to investigate pathogen evolution during transient epidemics and it might be useful to recall these alternative tools in the present work. If the question is "are virulent pathogens more likely to emerge in a market or in a farm?" (this seems to be the main motivation of the present work) you may also want to develop stochastic models to measure the probability of non-extinction of a virulent pathogen in different environments.

Reviewer #3 (Remarks to the Author):

I enjoyed the paper by Sheen et al. It considers an important problem: what is the impact of live bird markets on the evolution of pathogen virulence and transmission. The problem is tackled using a general, compartmental, model of infectious disease. This means that the results are applicable, and provide insight, beyond the focal study system. The findings indicate that the higher turnover rates of hosts and a reduced levels of decay of environmental pathogen associated with market environments can lead to selection for higher virulence. These results in themselves are not unexpected. The increase in turnover rate (μ) is similar to an increased death rate – which has been shown to select for increased virulence and transmission. The reduced level of decay of environmental pathogen leads to increased levels and persistence of free-living pathogen, and it has been previously shown that this can facilitate an increase in levels of direct transmission (and thereby virulence). Notwithstanding, the focus on farmed systems and the interpretation in terms of market interactions is new and interesting and will be of interest to a wide readership.

I would like to see this work published – but at present I feel that some major modifications are required before the work can be accepted. Some of these are related to the precision and clarity of the written work – and will be relatively easy to revise. However, I do also have some questions that could uncover potential issues with the model framework and evolutionary analysis. These will need to be checked and updated by the authors to ensure that the findings still hold, or may require the findings to be updated.

Major Issues:

Adaptive dynamics analysis

L204: We find the optimal virulence strategy by finding the virulence strategy with the largest R_0 in the system, i.e., following an adaptive dynamics approach.

While maximising R_0 is often equivalent to finding the ESS using adaptive dynamics – this is not always the case. An adaptive dynamics approach would consider the fitness of a mutant infection strain (a strain with a different ' α ') that attempts to invade system with an established resident strain (at equilibrium). The fitness can normally be determined as the largest eigenvalue of the ' μ mutant invasion matrix'. Once the fitness is determined the ESS occurs when the fitness gradient is zero, and it can be checked that the ESS is an evolutionary attractor (convergence and evolutionary stable). It could also then be checked that the adaptive dynamics criteria that determines the ESS is equivalent to maximising R_0 . If this is confirmed then using the R_0 maximisation approach is valid.

If the R_0 maximisation is valid, can you determine an explicit expression for the ESS by differentiating R_0 w.r.t α and setting this equal to zero? It may be that this is what you already do to produce your figures – but this was not clear to me.

Model formulation and description

L118: I think it would be good if you could justify the model choice. Why did you choose SEIR rather than SEI, SI, SIR etc. You mention on L159 'We assume that recovered poultry will never become susceptible or infectious again, which is a reasonable assumption due to the high antibody response to infection and relatively short life-cycle of the host (approx. one year)' but in the introduction you state the MDV, NDV, H5N1 are deadly pathogen and can lead to up to 100% mortality and led me to question whether the recovered class is applicable.

L126: The term for the production of environmental pathogen is $\sigma \cdot E_m$. I was confused as to why the pathogen was produced by exposed individuals – but not infected individuals. A more common way to model the production of environmental pathogen would be to have a rate of shedding, λ , from infected individuals ($\lambda \cdot I_m$) as in the classical work of Anderson and May 1981, Model G. The mathematical formulation, with production from class E_m , also contradicts with statement on L167 which states 'We model persistence and environmental transmission through mucus or faeces in the environment of an infectious chicken as follows: mucus and faeces are continually shed into the environmental reservoir for each infectious poultry in the system.'

L131: S_f - the number of susceptible poultry in farms - is in the model but I could not find an equation or explanation of the dynamics/density of S_f . I presume it is a constant. It would be worth justifying why only susceptible poultry can migrate from farms to markets.

L170: The manuscript states 'The reservoir decays at the rate of the average time period that an infectious poultry stays in the system (which accounts for the infectiousness period, migration rate out of the market, natural mortality, and disease-induced mortality rate), plus a delay due to the natural decay of the reservoir that is left after the infectious poultry is no longer in the system.' It is not clear what this means – either from the above text or Table 1. I think this is related to the term Ψ . If so I would like to see this defined explicitly in terms of the other model parameters.

L275: The manuscript states that 'We verify that markets can sustain the optimal virulence strategies of all tradeoff curves considered' and the abstract 'Additionally, more virulent strategies selected in markets can circulate solely within poultry located in markets'. I found this argument a little circular. We cannot examine the evolution of the pathogen in markets unless the pathogen can be supported (which means $R_0 > 1$ is a requirement for the evolutionary analysis).

Minor Issues:

I detail the sections/statements I found a difficult to read, or needed to read a few time to understand and some sections where I think precision could be improved. Please forgive me for being a pedant.

L45-52: These 2 sentences are long and difficult to read – Consider a rewrite for clarity?

L64-66: This sentence was not clear. Consider a rewrite for clarity?

L77-78: Consider a rewrite for clarity?

L85-89: This sentence was long and difficult to read – Consider a rewrite for clarity?

L101-113: I had to read this several times to work out the methods strategy. It was easier to understand after reading the whole manuscript – but maybe a rewrite for clarity would be helpful.

L128: Define $N_m = S_m + E_m + I_m + R_m$

L131: 'Hm is the number of units of reservoir in markets'. Units of reservoir is a strange term. Maybe units of pathogen in the environmental reservoir?

135: 'p sets whether transmission occurs in a density dependent or frequency dependent fashion'. But p is not in the model description shown in the equations L122-126.

L144-153: I felt this statement could be condensed.

L191: The manuscript states 'because it will simply raise the R0 of all virulence strategies by a constant, without modifying the shape of the tradeoff curve'. The shape of the trade-off curve is fixed for each model scenario so do you mean without modifying the position of the ESS?

L259: 'Higher virulent strategies will be selected when there is fast turnover (μ) and less cleaning (κ)' here higher virulence also means high transmission but this seems to contradict with the statement on the next line which says increases in μ will reduce transmission.

L260: 'As expected, increases in both μ and κ decrease the R0 of all virulence strategies; increases in both of these parameters reduce transmission'

L268: 'Increasing μ also decreases differences in R0 between the ESS and the maximum virulence strategy'. I'm not sure of the meaning or significance of this statement.

L281: 'Differences in the optimal virulence strategy in farms and markets will depend solely on differences in the parameters μ and κ (see "Methods: Characterizing Ecological Factors of Markets"). Because farms have relatively lower μ and relatively higher κ , markets will have higher optimal virulence strategies than farms (Figure 3)'. I agree, in principle, with this statement, but in the appendix you also highlight other parameters that will be different on farms. How do we know that the magnitude of virulence will not be affected by these 'other' parameter changes.

L619: 'However, of these parameters, only μ and κ will potentially change the shape of the tradeoff curve dependent on virulence.' These parameters do not effect the shape of the trade-off curve which is defined in L218,219.

Figure 3 – the scale shown does not have much contrast (it's almost all dark blue). As the figure is in colour figure could you use more colours to show the variation more clearly?

L648: Reference 10 – missing authors.

Point-by-Point Response to Reviews

The original reviewer comments are included here, followed by our response to each one in bold face:

General comments:

- **For clarity, we remove the “m” subscript signifying markets from all compartments and parameters since this is implicit.**

Reviewer: 1

- In this manuscript, the authors construct a mathematical model with the aim of investigating virulence evolution in rural markets and farms. Overall, this is an interesting question – “what ecological factors in rural settings shape virulence evolution” - that has yet to be addressed with mathematical modelling.

Thank you for this comment. We hope our modeling work can shed light on this important question, as well as open a door towards future modeling and empirical efforts to extend our results.

- I believe the authors have a bit more work to do. My major concern is with the evolutionary analysis. On line 203 the authors reference ‘The Hitchhiker’s Guide to Adaptive Dynamics’. In this reference the evolutionary stable strategy is found via optimizing invasion fitness. But what the authors have done in this manuscript is optimize R_0 . I suggest the authors calculate invasion fitness in this study, or explain why optimizing R_0 is the better approach. Please see ‘Beyond R_0 Maximisation: On Pathogen Evolution and Environmental Dimensions’ by Lion and Metz. <https://pubmed.ncbi.nlm.nih.gov/29665966/> for clarification.

Thank you for this comment. In this resubmission, we develop a formal expression for the invasion fitness, $\mathcal{R}_i(\alpha_r, \alpha_m)$ ($s_r(m)$ in ‘The Hitchhiker’s Guide to Adaptive Dynamics’), to prove that for our model (Appendix 2), as well as for a model with differential migration of infectious poultry (Appendix 4), the virulence strategy that maximizes \mathcal{R}_0 compared to all other virulence strategies is the strategy that can invade all other strategies; and is evolutionarily and convergent stable. Thus, the \mathcal{R}_0 optimization approach is sufficient for a valid adaptive dynamics analysis. From another perspective, because the effect of the environment can be characterized by a single resource, namely the equilibrium of susceptible individuals, finding the virulence strategy that maximizes \mathcal{R}_0 , resulting in the lowest susceptible equilibrium, is sufficient for predictions of virulence evolution (Lion and Metz 2018).

- As the authors revisit the mathematical analysis of this model, I think it would be interesting to explore a “migration out of markets” that is dependent of disease status. I would think that diseased birds (and since they are not vaccinated, it is likely they would look “sick”) would “migrate” out of markets at a slower rate.. since they look less delicious. This would be interesting since this could be a negative selective pressure on virulence evolution. The lifespan of infected birds would be longer for “sick” birds, and this would select for less virulent varieties.

This is an interesting point, and we now include a sensitivity analysis where poultry in the I compartment migrate at a 50% slower rate out of the markets compared to all other compartments (results in Table S3 in Appendix 3, derivation of model and validity of R_0 maximization approach in Appendix 4). We find that although there are still decreases in the global ESS when decreasing m, the maximum decreases will be somewhat smaller when there is differential emigration. We now report these results, as well as include a supplemental table (Table S3, Appendix 3). The results of this sensitivity test make sense, as longer durations of infectious poultry in markets may reduce the reductions in virulence we see when slowing the emigration rate, as you note.

Whether “sicker” birds actually emigrate out of markets at a slower rate is unclear. Although “sicker” looking birds may be less likely to be sold out of markets, slowing the emigration rate, sellers may also remove these “sicker” birds from mixing with other poultry to avoid spread of the pathogen, quickening the emigration rate. However, across both our main results and this sensitivity test, we find decreases in the global ESS whether or not emigration is different for infectious poultry compared to all other poultry.

- There is a typo in the model presented in the main text. There is no ‘p’ term to toggle between FD and DD transmission. Some motivation for FD vs DD transmission would be helpful. When comparing Newcastle, Marek’s and HPAI, are one of those viruses assumed to spread via FD transmission?

Thank you for catching this. We have chosen to eliminate the ‘p’ term to toggle between frequency-dependent and density-dependent transmission from all model diagrams and equations until Appendix 1, where we introduce a generalized version of our model in the text in order to derive the R_0 both when transmission is frequency-dependent (p=1) or density-dependent (p=0). We have also added additional motivation for modeling density-dependent transmission in lines 175-178: “In this article, we model density dependent transmission: assuming the area of the

markets modeled is fixed, a larger population size will increase the contact rate of poultry; thus, the transmission rate is density-dependent.”

- It would also be helpful if the authors could elaborate on what the reservoir, ‘H_m’, represents biologically. At first, I assumed this was a free living “viral” population. You mention on line 168 that “mucus and faeces are continually shed into the environmental reservoir for each infectious poultry in the system.” But, there is no contribution to this ‘H_m’ compartment from the infectious class (only the exposed class). So, it is unclear what this compartment represents.

Thank you for this suggestion. We have changed the way we model the environmental reservoir so that now there is only contribution to the H compartment from the infectious class. Biologically, the H compartment is made up of environmental pathogen particles. In our model, when $\varepsilon = 1$, the units of H can be understood as some amount of environmental pathogen particles that have a transmission rate equivalent to one infectious poultry.

- I think if the authors make these changes, this can be a very interesting and worthwhile paper.

Thank you, we hope our work can spark interest for this important area of research.

Reviewer: 2

- This study explores the effect of rural markets on the evolution of virulence of poultry pathogens. The spread of the pathogen in a market is governed by the influx and the outflux of poultry and by the transmission between birds but also from an environmental compartment. The pathogen population is assumed to be unaffected by evolution going on in other farms or other markets. The analysis is based on the optimisation of R₀: the ESS virulence is the one maximizing R₀ but this value may vary with the parameters that govern the life cycle in the market. The authors conclude that ESS virulence increases with fast turnover and less cleaning.

Thank you for this summary of our work.

- I have several major issues with this work:
- (1) Some assumptions of the model are unclear. H_m is supposed to be “units of reservoir in markets” but this probably means the density of pathogen propagules in the environment. Why are these propagules produced by E_m at a rate σ (E_m are supposed to be not infectious)? Also, why do you introduce β_m as a function of H_m? It would be simpler to model the density H_m to be a function of infectious hosts

(i.e. the first term of the ast equation could be $\beta^E_{m} I_m$, where the E would stand for the production of propagules in the environment. In addition, the parameter p is not well defined. I understand it affects R0 but it should be introduced in the EDO model too.

Thank you for this suggestion. We have changed the way we model the environmental reservoir so that now there is only contribution to the H compartment from the infectious class. Biologically, the H compartment is made up of environmental pathogen particles. In our model, when $\varepsilon = 1$, the units of H can be understood as some amount of environmental pathogen particles that have a transmission rate equivalent to one infectious poultry.

We have now removed the ‘p’ parameter from the model introduced in the text as to avoid confusion since we solely model density-dependent transmission. We more clearly defined the ‘p’ parameter in lines 623-625 of the supplement where we introduce a generalized version of the model in the text in order to derive R_0 generally across both frequency-dependent (p=1) and density-dependent (p=0) conditions: “We derive the R_0 of a generalized version of our model that includes the potential for frequency-dependent transmission by including the parameter, p, where p=0 makes transmission density-dependent and p=1 makes transmission frequency-dependent”

- (2) The prediction that increased mortality selects for higher ESS virulence is a very classic prediction (a prediction that has been put to the test experimentally some time ago, Ebert and Mangin 1997 in Evolution). Increasing turnover is the same effect and it should be discussed.

Thank you for this suggestion. In addition to mentioning that the prediction of increased mortality will select for higher virulence has been tested for Marek’s disease virus in the introduction, we also note in the discussion that our result echoes this classic result is now included in lines 427-429: “The former echoes a classic result in the literature of virulence evolution where higher host mortality will select for increased virulence in the novel context of live poultry markets (Anderson and May 1982, Atkins et al. 2013, Nidelet et al. 2009, Sasaki and Iwasa 1991).”

- (3) The prediction that less cleaning should select for more virulence contrast with previous analysis. In ref [1] the authors show that increased cleaning is expected to select for higher virulence. Why the difference? Probably in the underlying assumption of the demography. The present work assumes the population has reached a stable endemic equilibrium before the introduction of a new mutant. In ref [1] the demography fluctuates

periodically. None of these assumptions are likely to be relevant in a market so it is difficult to make predictions about the effect of cleaning.

The reason that there is a difference may be due to the assumption of an intercohort period in Ref [1], where at the end of each cohort both susceptible and infectious poultry are removed (all-in-all-out), and the number of infectious poultry in the new cohort will depend on the proportion of environmental pathogen particles that survive the intercohort period:

- **The cost of the transmission-virulence tradeoff comes from virulence decreasing the lifespan of an infectious poultry (seen in the ν parameter of the dI_1/dt and dI_2/dt equations of model (3) of Ref [1], as well as in the δ parameter of the dI/dt equation in our model).**
- **Because cleaning decreases the proportion of environmental pathogen particles that survive the intercohort period, cleaning decreases the number of infectious poultry in the new cohort period (see final two equations of reduced model (3) of Ref[1], noting that $I_{1(2)}(nT^+)$ depends on $a_{1(2)}$ with the cleaning parameter, $\gamma_{1(2)}$).**
- **Thus, cleaning raises the global ESS since the relative cost of high virulence is low because the lifespan of infectious poultry is shortened.**
- **In our model, while cleaning also decreases the lifespan of environmental pathogen particles, it does not decrease the lifespan of infectious poultry, which again, is where a cost due to virulence arises. In other words, cleaning does not raise the global ESS in our model because the relative cost of high virulence compared to lifespan of infectious poultry is unaffected by cleaning.**

All-in-all-out dynamics may be less common in the rural live-poultry markets we model. Sellers often receive a new cohort of poultry to sell before all of the poultry of the older cohort are sold. Additionally, even if some sellers may practice all-in-all-out cohort practices, the number of infectious poultry of each new cohort of one seller may depend not only on how well the seller cleans the particles from their own stall, but also on the prevalence in markets. Thus, the continuous dynamics we model may be more appropriate for modeling rural market systems than the “all-in-all-out” intercohort dynamics used to model industrialized poultry systems in Ref [1]. This assumption is now discussed in full in the Discussion in lines 450-459.

In our model, where cleaning does not affect the lifespan of an infectious poultry, the intuition behind cleaning selecting for lower virulence is shown when viewing the \mathcal{R}_0 of the system:

$$R_0 = \left(\frac{m_f S_f}{\mu + m} \right) \left(\frac{\sigma}{\sigma + \mu + m} \right) \left(\frac{1}{m + \gamma + \delta + \mu} \right) \beta \left[1 + \frac{\epsilon \lambda}{\kappa \psi} \right]$$

As cleaning increases, the second term within the brackets, which when multiplied with β is the environmental transmission benefit, will decrease to zero. Thus, the higher virulence strategies that stand to gain the most from environmental transmission when there is no cleaning (since λ is an increasing function of virulence) will suffer the greatest reduction in transmission benefits when there is cleaning. Without this added environmental transmission benefit, higher virulence strategies are less valuable from the pathogen's perspective, and lower virulence strategies are selected.

- (4) More generally, I am not sure the looking for the ESS is most relevant approach to this question as we may want to determine the fate of a mutant during a transient epidemic. Different tools have been used to investigate pathogen evolution during transient epidemics and it might be useful to recall these alternative tools in the present work. If the question is “are virulent pathogens more likely to emerge in a market or in a farm?” (this seems to be the main motivation of the present work) you may also want to develop stochastic models to measure the probability of non-extinction of a virulent pathogen in different environments.

Thank you for this comment. Indeed, one of the major limits of the ESS approach we adopt is that it does not account for transience in a population, nor stochastic dynamics. However, we feel that it is useful to explore intuition as to the outcome within equilibril settings as a foundation from which to build an understanding of transient dynamics.

Reviewer: 3

- I enjoyed the paper by Sheen et al. It considers an important problem: what is the impact of live bird markets on the evolution of pathogen virulence and transmission. The problem is tackled using a general, compartmental, model of infectious disease. This means that the results are applicable, and provide insight, beyond the focal study system. The findings indicate that the higher turnover rates of hosts and a reduced levels of decay of environmental pathogen associated with market environments can lead to selection for higher virulence. These results in themselves are not unexpected. The increase in turnover rate (mm) is similar to an increased death rate – which has been shown to select for increased virulence and transmission. The reduced level of decay of environmental pathogen leads to increased levels and persistence of free-living pathogen, and it has been previously shown that this can facilitate an increase in levels of direct transmission (and thereby virulence). Notwithstanding, the focus on farmed systems and the interpretation

in terms of market interactions is new and interesting and will be of interest to a wide readership.

Thank you for your comment. We hope that our work can shed light on this important area of research.

- I would like to see this work published – but at present I feel that some major modifications are required before the work can be accepted. Some of these are related to the precision and clarity of the written work – and will be relatively easy to revise. However, I do also have some questions that could uncover potential issues with the model framework and evolutionary analysis. These will need to be checked and updated by the authors to ensure that the findings still hold, or may require the findings to be updated.

Thank you, we hope that the following responses will adequately address your concerns.

- Major Issues:
- Adaptive dynamics analysis
- L204: We find the optimal virulence strategy by finding the virulence strategy with the largest R_0 in the system, i.e., following an adaptive dynamics approach.

While maximising R_0 is often equivalent to finding the ESS using adaptive dynamics – this is not always the case. An adaptive dynamics approach would consider the fitness of a mutant infection strain (a strain with a different ‘alpha’) that attempts to invade system with an established resident strain (at equilibrium). The fitness can normally be determined as the largest eigenvalue of the ‘mutant invasion matrix’. Once the fitness is determined the ESS occurs when the fitness gradient is zero, and it can be checked that the ESS is an evolutionary attractor (convergence and evolutionary stable). It could also then be checked that the adaptive dynamics criteria that determines the ESS is equivalent to maximising R_0 . If this is confirmed then using the R_0 maximisation approach is valid.

If the R_0 maximisation is valid, can you determine an explicit expression for the ESS by differentiating R_0 w.r.t alpha and setting this equal to zero? It may be that this is what you already do to produce your figures – but this was not clear to me.

Thank you for this comment. We have now addressed the question of whether R_0 maximization is a valid approach for an adaptive dynamics analysis of our model: copied from the above response to Reviewer 1:

“In this resubmission, we develop a formal expression for the invasion fitness, $\mathcal{R}_0(\alpha_r, \alpha_m)$ ($s_r(m)$ in ‘The Hitchhiker’s Guide to Adaptive Dynamics’), to prove that for our model (Appendix 2), as well as for a model with differential migration of infectious poultry (Appendix 4), the virulence strategy that maximizes \mathcal{R}_0 compared to all other virulence strategies is the strategy that can invade all other strategies; and is evolutionarily and convergent stable. Thus, the \mathcal{R}_0 optimization approach is sufficient for a valid adaptive dynamics analysis. From another perspective, because the effect of the environment can be characterized by a single resource, namely the equilibrium of susceptible individuals, finding the virulence strategy that maximizes \mathcal{R}_0 is sufficient for predictions of virulence evolution (Lion and Metz 2018).”

Thank you for the suggestion of determining an explicit expression for the ESS. We’ve tried to derive an analytical solution, but the algebra is unfortunately intractable. Instead, we create our figures by scanning through all virulence strategies from $\alpha = 1$ to 1000 with step size = 1, finding the α that maximizes \mathcal{R}_0 .

- Model formulation and description
- L118: I think it would be good if you could justify the model choice. Why did you choose SEIR rather than SEI, SI, SIR etc. You mention on L159 ‘We assume that recovered poultry will never become susceptible or infectious again, which is a reasonable assumption due to the high antibody response to infection and relatively short life-cycle of the host (approx. one year)’ but in the introduction you state the MDV, NDV, H5N1 are deadly pathogen and can lead to up to 100% mortality and led me to question whether the recovered class is applicable.

Thank you for this suggestion. We apologize, as we should have clarified that although some strains of these viruses can have up to 100% mortality, this is not true for all strains (e.g. Class I viruses of NDV, and the low pathogenic avian influenza viruses (LPAI)). We have now clarified this in lines 33-35: “Since their discovery, all three pathogens have been shown to circulate worldwide, and the most virulent strains of each have been reported to kill up to 100% of poultry flocks.”

We also include a statement to motivate our SEIR modeling choice in lines 141-146: “The SEI model framework has been selected as an appropriate model for poultry pathogens such as Newcastle disease virus and Marek’s disease virus.^{29,30} Although in the current model, recovered individuals do not affect infection dynamics, we extend the SEI model to include a recovered class to model the possibility of

recovered individuals affecting infection dynamics in future extensions (e.g. when transmission is frequency-dependent)."

- L126: The term for the production of environmental pathogen is $\sigma \cdot E_m$. I was confused as to why the pathogen was produced by exposed individuals – but not infected individuals. A more common way to model the production of environmental pathogen would be to have a rate of shedding, λ , from infected individuals ($\lambda \cdot I_m$) as in the classical work of Anderson and May 1981, Model G. The mathematical formulation, with production from class E_m , also contradicts with statement on L167 which states ‘We model persistence and environmental transmission through mucus or faeces in the environment of an infectious chicken as follows: mucus and faeces are continually shed into the environmental reservoir for each infectious poultry in the system.’

Thank you for this comment. We have changed the way we model the environmental reservoir so that now there is only contribution to the H compartment from the infectious class. Biologically, the H compartment is made up of environmental pathogen particles. In our model, when $\varepsilon = 1$, the units of H can be understood as the amount of environmental pathogen particles that have a transmission rate equivalent to one infectious poultry.

- L131: S_f - the number of susceptible poultry in farms - is in the model but I could not find an equation or explanation of the dynamics/density of S_f . I presume it is a constant. It would be worth justifying why only susceptible poultry can migrate from farms to markets.

Thank you for this comment. Yes, we assume S_f is a constant of the number of susceptible poultry in farms. We have included a clarification on this point in lines 162-163: “ S_f is the number of susceptible poultry in farms which we assume is a constant.”

Conceptually, less symptomatic poultry (i.e. non-infected poultry) may be more likely to be sold to middlemen, and consequently to traders. Another reason that we model that solely susceptible poultry can migrate from farms to markets is that our adaptive dynamics approach currently cannot account for dynamics when infected poultry migrate into the system since there is no longer a guaranteed unique endemic equilibrium for each virulence strategy, since it is no longer a sub-model of the transmission model introduced in Shuai and van den Driessche (2011). Future work may seek to account for situations when infected poultry can migrate into the system. We have now added a section in the main text concerning this point in lines 279-284: “Only susceptible poultry from farms are allowed to migrate to markets.

Conceptually, more symptomatic poultry are less likely to be sold to middlemen and traders due to their diseased appearance. Additionally, our adaptive dynamics analysis cannot account for when infected poultry from farms are able to migrate to markets, since in this case, it has not yet been proven that there is a unique endemic equilibrium for each virulence strategy. Future studies may aim to account for cases where infectious poultry can migrate to markets.”

- L170: The manuscript states ‘The reservoir decays at the rate of the average time period that an infectious poultry stays in the system (which accounts for the infectiousness period, migration rate out of the market, natural mortality, and disease-induced mortality rate), plus a delay due to the natural decay of the reservoir that is left after the infectious poultry is no longer in the system.’ It is not clear what this means – either from the above text or Table 1. I think this is related to the term Ψ . If so I would like to see this defined explicitly in terms of the other model parameters.

Thank you for this suggestion. Since we have switched models, Ψ is much simpler, as it is simply the natural decay of the environmental pathogen particles in the reservoir, H (Table S1).

- L275: The manuscript states that ‘We verify that markets can sustain the optimal virulence strategies of all tradeoff curves considered’ and the abstract ‘Additionally, more virulent strategies selected in markets can circulate solely within poultry located in markets’. I found this argument a little circular. We cannot examine the evolution of the pathogen in markets unless the pathogen can be supported (which means $R_0 > 1$ is a requirement for the evolutionary analysis).

Thank you for this comment. We have now removed this superfluous result to focus our results on the adaptive dynamics analysis results.

- Minor Issues:
- I detail the sections/statements I found a difficult to read, or needed to read a few time to understand and some sections where I think precision could be improved. Please forgive me for being a pedant.

Thank you for these catches, suggestions, and comments. We truly appreciate them.

- L45-52: These 2 sentences are long and difficult to read – Consider a rewrite for clarity?

Thank you for this suggestion. These sentences are now rewritten in lines 45-49 as: “When higher virulence is positively associated with increasing rates of

transmission, a tradeoff between transmission and mortality emerges. Although there is a benefit to increasing virulence due to increasing transmission, there is also a cost of shortening the lifespan of the host, and consequently the infectious period of the pathogen. In the past, researchers have identified several husbandry practices that either increase the transmission benefits of virulence, or reduce the mortality costs of virulence, ultimately selecting for increased virulence.”

- L64-66: This sentence was not clear. Consider a rewrite for clarity?

Thank you for this suggestion. This sentence is now rewritten in lines 69-71 as: “Because there appears to be uninterrupted pathogen transmission in markets (rather than extinction/reintroduction) markets can introduce selection pressures on the pathogens.”

- L77-78: Consider a rewrite for clarity?

Thank you for this suggestion. This sentence is now rewritten in lines 85-87 as: “However, while many lines of evidence suggest that markets might select for increases in virulence, these hypotheses have yet to be rigorously examined. The magnitude of these increases also remains unclear.”

- L85-89: This sentence was long and difficult to read – Consider a rewrite for clarity?

Thank you for this suggestion. This sentence is now rewritten in lines 96-97 as: “Thus, we use the R_0 expression to titrate the impact of turnover rate and environmental persistence on the global ESS.”

- L101-113: I had to read this several times to work out the methods strategy. It was easier to understand after reading the whole manuscript – but maybe a rewrite for clarity would be helpful.

Thank you for this suggestion. This paragraph is now rewritten in lines 113-124 as: “Our aim is to characterize the evolution of the optimal level of virulence, or host mortality rate, in contexts where there is a relationship between virulence and transmission (Figure 1), when transmission occurs in markets. To do this, we perform an adaptive dynamics analysis to find how the evolutionarily stable strategy (ESS) changes under market conditions. We first derive and analyze the R_0 of a model designed to capture the market conditions of faster turnover rates and higher environmental persistence of viral particles in markets (“Transmission Assumptions in Markets”). We then define our assumptions of the

transmission-mortality tradeoff curve (“Virulence Assumptions.”) Finally, we vary the turnover rate and persistence parameters from farm conditions to market conditions and evaluate how the ESS changes relative to farms (“Characterizing Ecological Factors of Markets”). The code used to generate these results, as well as the results themselves, are openly available at <http://www.github.com/jsheen/marketVirEvol>.”

- L128: Define $N_m = S_m + E_m + I_m + R_m$

Thank you for this suggestion. This is now included in line 158.

- L131: ‘ H_m is the number of units of reservoir in markets’. Units of reservoir is a strange term. Maybe units of pathogen in the environmental reservoir?

Thank you for this suggestion. This is now rewritten in line 161: “ H is the units of pathogen particles in the environmental reservoir.”

- 135: ‘ p sets whether transmission occurs in a density dependent or frequency dependent fashion’. But p is not in the model description shown in the equations L122-126.

Thank you for this catch. This is now fixed to include p .

- L144-153: I felt this statement could be condensed.

Thank you for this suggestion. We felt that the subsequent part of the paragraph could also be condensed, and these sentences are now rewritten in lines 181-185 as: “Although poultry are primarily located in live-bird markets, our model accounts for times when poultry are transported by middlemen to traders, as well as if they are housed anywhere on off-days of markets. As long as poultry are continuously mixing in these places, our model is appropriate for modeling pathogen transmission (e.g., settings like Antananrivo, Madagascar where markets are open year-round).”

- L191: The manuscript states ‘because it will simply raise the R_0 of all virulence strategies by a constant, without modifying the shape of the tradeoff curve’. The shape of the trade-off curve is fixed for each model scenario so do you mean without modifying the position of the ESS?

Thank you for this catch. Yes, this is exactly what we meant. We have now clarified this sentence in lines 234-237: “First, because β can be factored out of the equation, a higher contact rate in markets relative to farms will simply raise the R_0 of all

virulence strategies by a constant, and thus will not change the position of the global ESS.”

- L259: ‘Higher virulent strategies will be selected when there is fast turnover (m) and less cleaning (κ)’ here higher virulence also means high transmission but this seems to contradict with the statement on the next line which says increases in m will reduce transmission.

Thank you for this catch. We simplify this statement to solely say that increases in m (fast turnover) and increases in κ (fast cleaning) both reduce the R_0 for all virulence strategies. We had mistakenly originally conflated R_0 with transmission, but this is not correct. This is now corrected in line 353: “As expected, increases in both m and κ decrease the R_0 of all virulence strategies”

Again, although the R_0 of all virulence strategies is reduced, virulence may still be selected to be higher due to fast turnover (m) and reduced cleaning (κ).

- L260: ‘As expected, increases in both m and κ decrease the R_0 of all virulence strategies; increases in both of these parameters reduce transmission’

Thank you for this catch. We have now addressed this comment in the response directed above this one.

- L268: ‘Increasing m also decreases differences in R_0 between the ESS and the maximum virulence strategy’. I’m not sure of the meaning or significance of this statement.

Thank you for this comment. We have now eliminated this sentence, since it is irrelevant to the main adaptive dynamics results.

- L281: ‘Differences in the optimal virulence strategy in farms and markets will depend solely on differences in the parameters m and κ (see “Methods: Characterizing Ecological Factors of Markets”). Because farms have relatively lower m and relatively higher κ , markets will have higher optimal virulence strategies than farms (Figure 3).’ I agree, in principle, with this statement, but in the appendix you also highlight other parameters that will be different on farms. How do we know that the magnitude of virulence will not be affected by these ‘other’ parameter changes.

Thank you for this comment. Of the first five characteristics listed, we have justified, according to our model, that solely m and κ are hypothesized to have an

effect on the selection of the ESS virulence in “Justification of comparison of selection pressures in markets vs. farms” in Appendix 3. The last characteristic, “Local vs. global transmission,” is not considered in our model, but is discussed in depth in the Discussion of our paper. We find that, if anything, this may actually further increase differences in the ESS between farms and markets. We have updated the caption of Figure S1 in lines 615-620 to include this information: “According to our model, of the first five characteristics, m and κ are hypothesized to have an effect on the selection of the ESS virulence (see “Justification of comparison of selection pressures in markets vs. farms” in Appendix 3). The last characteristic, Local vs. global transmission, is not captured by our model, but may further create differences between the ESS virulence in farms vs. markets (see Discussion).”

- L619: ‘However, of these parameters, only m and κ will potentially change the shape of the tradeoff curve dependent on virulence.’ These parameters do not effect the shape of the trade-off curve which is defined in L218,219.

Thank you again for this catch of our typo. We have now rewritten this, and condensed it, in lines 970-973: “However, of these parameters, only m and κ will potentially change the shape of the fitness landscape, and thus change the position of the global ESS: b is and m_f are not included in the R_0 expression, and changes in m_f , S_f , and β scale R_0 and would not change the position of the ESS.”

- Figure 3 – the scale shown does not have much contrast (it’s almost all dark blue). As the figure is in colour figure could you use more colours to show the variation more clearly?

Thank you for this suggestion. We have now changed the figure in lines 559-569 to include more colors to show the variation more clearly.

- L648: Reference 10 – missing authors.

Thank you for this catch. We have now included the authors.

REVIEWER COMMENTS

Reviewer #2 (Remarks to the Author):

The revisions of the manuscript clarified many points but I am still unclear about the novelty and the validity of some of the predictions. The main effects discussed by the authors are the effect of the turnover rate (m) and the effect of the expected lifespan of pathogen particles ($1/\kappa$) on the ESS virulence.

First, as pointed out by other referees, the increase of the ESS virulence with higher turnover rate is not new. This has been discussed in several previous studies. Some experimental studies even tried to test this prediction (Ebert&Mangin 1997 Evolution). They failed to validate this prediction which led to further theory (Gandonetal 2001 Evolution). It would be interesting to refer to these earlier studies because they question the validity of this classical prediction.

Second, the discussion on the effect of κ is not entirely satisfying. The authors suggest that the discrepancy with earlier work (reference [1]) stems from different assumptions in the temporal dynamics of the within the farm. But (1) the authors do not demonstrate that the qualitatively different prediction is due to this assumption and (2) the previous study was tailored to disease spread in poultry so it is likely to be very relevant for the present work. So I am still unclear about what to expect if one increases the mortality rate of propagules on the ESS virulence. You may want to refer to the reanalysis of the "curse of the pharaoh hypothesis" presented in Lion&Gandon Evolution 2022. The mortality rate of the propagule (similar to κ) is expected to have no effect on ESS virulence at the endemic equilibrium (the situation analysed in the present study) but could select for lower or higher ESS virulence in a periodically fluctuating environment (perhaps closer to reference [1]). I am unclear about why the effect of this parameter is so sensitive to the underlying assumptions of the model but I feel it is important to explore this further because it affects one of the key prediction of the present study.

Minor comment: you should reword the legend of figure 3 because it sounds like κ measures environmental persistence (instead, the expected persistence is $1/\kappa$).

Reviewer #3 (Remarks to the Author):

The resubmitted manuscript 'Markets can select for and maintain highly virulent poultry pathogens' is a comprehensive revision of the original paper. The authors have taken on board all my original comments and provided a clear and detailed response to these comments and updated the manuscript with respect to my concerns. I commend the authors on their thorough revision. I enjoyed reading the revised manuscript and believe it will be of interest to the wide readership of Nature Comms. I have no further comments and recommend that the article is published.

Point-by-Point Response to Reviews

The original reviewer comments are included here, followed by our response to each one in bold face:

Reviewer: 2

- The revisions of the manuscript clarified many points but I am still unclear about the novelty and the validity of some of the predictions. The main effects discussed by the authors are the effect of the turnover rate (m) and the effect of the expected lifespan of pathogen particles ($1/\kappa$) on the ESS virulence.

Thank you for this comment, and for your helpful comments and suggestions throughout the review process. We hope we have clarified below and in the main text our points to fully address your concerns.

- First, as pointed out by other referees, the increase of the ESS virulence with higher turnover rate is not new. This has been discussed in several previous studies. Some experimental studies even tried to test this prediction (Ebert&Mangin 1997 Evolution). They failed to validate this prediction which led to further theory (Gandonetal 2001 Evolution). It would be interesting to refer to these earlier studies because they question the validity of this classical prediction.

Thank you for this comment. With regard to the novelty of our results, although increased background mortality (equivalent to turnover rate) has previously been identified as a factor that can drive the evolution of virulence, how this will play out in the context of live-poultry markets, which reflect significant ecological differences, has yet to be evaluated. And although in some experimental studies the higher turnover rate leading to increased virulence failed to be validated, higher turnover rate for industrialized poultry populations has been suggested as a possible cause for the historical increase of virulence in the Marek's disease virus (Atkins et al. 2013, Rozins and Day 2017) which highlights how valuable the theory is in informing the evolution of poultry viruses. Whether the prediction holds for live-poultry markets is hard to intuit due to the ecological differences of markets compared to industrialized farms, such as the continuous inflows and outflows, increased environmental transmission, and uncertainty around the exact turnover rate. By parameterizing our model with real-world data collected from live-poultry markets, we were able to develop predictions for how turnover rates specific to this setting should affect the evolution of virulence.

The valuable experiment and theory by Ebert and Mangin (1997) and Gandon et al. (2001), respectively, demonstrates that when hosts have many coinfections there

may be higher within-host competition selecting for higher virulence, and thus that higher background host mortality may select for lower virulence since it will reduce the force of infection of these multiple pathogens. Indeed, this may be especially important in future extensions of our analyses, since poultry viruses such as Newcastle disease virus are known to co-infect hosts with many other pathogens (Panyako et al. 2023). We now reference this important area of future research, specifically whether the higher background host mortality of live-poultry markets would reduce the force of infection of multiple pathogens enough to select for lower virulence, in our Discussion in lines 324 to 333:

“Our analysis indicates that the rapid turnover rate in live-poultry markets will select for higher virulence, but additional factors, such as multiple infections, may affect our results, and remain an important area of future research. When hosts have multiple infections, there is higher within-host competition, which selects for higher virulence (Ebert and Mangin 1997, Gandon et al. 2001). When there is higher background host-mortality, within-host competition is potentially lessened since there is a lower force of infection, and lower virulence may be selected. Thus, the presence of multiple infections could affect our prediction that rapid turnover, and higher background host mortality, will select for increased virulence. Because poultry are often co-infected by many pathogens (Panyako et al. 2023), this possibility should be considered in future research, specifically whether the higher background host mortality in live-poultry markets would reduce the force of infection of multiple pathogens enough to select for lower virulence.”

- Second, the discussion on the effect of kappa is not entirely satisfying. The authors suggest that the discrepancy with earlier work (reference [1]) stems from different assumptions in the temporal dynamics of the within the farm. But (1) the authors do not demonstrate that the qualitatively different prediction is due to this assumption and (2) the previous study was tailored to disease spread in poultry so it is likely to be very relevant for the present work. So I am still unclear about what to expect if one increases the mortality rate of propagules on the ESS virulence. You may want to refer to the reanalysis of the "curse of the pharaoh hypothesis" presented in Lion&Gandon Evolution 2022. The mortality rate of the propagule (similar to kappa) is expected to have no effect on ESS virulence at the endemic equilibrium (the situation analysed in the present study) but could select for lower or higher ESS virulence in a periodically fluctuating environment (perhaps closer to reference [1]). I am unclear about why the effect of this parameter is so sensitive to the underlying assumptions of the model but I feel it is important to explore this further because it affects one of the key prediction of the present study.

Thank you for this comment. In order to further address this issue, we created a variation of our model that assumes distinct cohort and intercohort periods, where all poultry of one cohort are replaced by another cohort of poultry, and cleaning occurs solely during the intercohort period, as in Ref[1]. In this variation of our model, for an example tradeoff curve, we recover the result of Ref[1] that increased cleaning will select for an increased global ESS (Figure S1 below). Importantly, if we instead use the main model of our paper, we find that increased cleaning will instead decrease the ESS. These results lend further evidence that whether cleaning selects for increased or decreased virulence depends on whether there are distinct cohorts of poultry with cleaning between cohorts, since cleaning will directly affect the relative costs of virulence. By contrast, if cohorts overlap, cleaning will tend to reduce virulence, as presented in our main results. These points are now described in full in Appendix 6. They are also referenced in the results, as well as in the Discussion in lines 329-331:

“Our sensitivity analysis suggests that if we assume this dynamic, with cleaning solely during the intercohort period, cleaning will increase the global ESS (Appendix 6).”

Responding to point (2), although Ref [1] models disease spread in poultry, it does so in industrialized populations, which is different from the rural live poultry markets that we model. For rural live poultry markets, it may be more appropriate to use the model we present in our main results, since rural live-poultry markets, which have overlapping, partial replacement of cohorts from many different sellers. This is discussed in lines 331 to 337 in the main text:

“However, these all-in-all-out dynamics may be less common in the rural live-poultry markets we model. Sellers often receive a new cohort of poultry to sell before all of the poultry of the older cohort are sold. Additionally, even if some sellers may practice all-in-all-out cohort practices, the number of infectious poultry of each new cohort of one seller may depend not only on how well the seller cleans the particles from their own stall, but also on the prevalence in markets. Thus, the continuous dynamics we model may be more appropriate for modeling rural market systems.”

Figure S1: Pairwise invasibility plots with discrete cohorts showing success (black, >2 infected chickens at the end of the simulation) and failure (white, <1 infected chickens) of invasion by the strategy on the y axis (left column), extended to illustrate resident dynamics (right column, gray areas indicate resident virus persistence alongside invaders with >2 infected chicken at the end of the simulation) in a context of no cleaning (top row, $\kappa = 0$) and high cleaning (bottom row, a proportion $\kappa = 0.95$ of viral particles cleaned between cohorts). Cleaning selects for an increase in the evolutionary stable virulence strategy. Each cohort generation duration, $T = 90$ days. The transmission-mortality tradeoff parameters used were $c_1 = 1/2300$, $c_2 = 0.45$, $\Phi = 10$. Other parameters used were $\sigma = 1/5$ days, $\gamma = 1/5$ days, $\mu = 1/365$ days, $\psi = 1/5$ days, $m = 1/5.5$ days, $m_f = 0.1 / 120$ days, $S_f = 1$ million. For each cell, the resident strategy was run for 1000 cohort generations, a single infected poultry with the invading virulence strategy was introduced and the simulation was run further for 1000, 10000, or 200,000 cohort generations, depending on whether there was clear growth or eradication of the invading strategy.

- Minor comment: you should reword the legend of figure 3 because it sounds like kappa measures environmental persistence (instead, the expected persistence is $1/\text{kappa}$).

Thank you for this catch, we have now corrected the figure legend.

Reviewer: 3

- The resubmitted manuscript ‘Markets can select for and maintain highly virulent poultry pathogens’ is a comprehensive revision of the original paper. The authors have taken on board all my original comments and provided a clear and detailed response to these comments and updated the manuscript with respect to my concerns. I commend the authors on their thorough revision. I enjoyed reading the revised manuscript and believe it will be of interest to the wide readership of Nature Comms. I have no further comments and recommend that the article is published.

Thank you for your helpful comments and suggestions during the reviewing process.